# LAND: Lung and Nodule Diffusion for 3D Chest CT Synthesis with Anatomical Guidance

## Abstract

This work introduces a new latent diffusion model to generate high-quality 3D chest CT scans conditioned on 3D anatomical masks. The method synthesizes volumetric images of size $256 \times 256 \times 256$ at 1 mm isotropic resolution using a single mid-range GPU, significantly lowering the computational cost compared to existing approaches. The conditioning masks delineate lung and nodule regions, enabling precise control over the output anatomical features. Experimental results demonstrate that conditioning solely on nodule masks leads to anatomically incorrect outputs, highlighting the importance of incorporating global lung structure for accurate conditional synthesis. The proposed approach supports the generation of diverse CT volumes with and without lung nodules of varying attributes, providing a valuable tool for training AI models or healthcare professionals. Code for LAND is available at: https://github.com/anonymous/LAND-3DCT.

## 1 Introduction

Deep learning in medical imaging is hindered by the scarcity of large, diverse datasets, constrained by privacy concerns, costs, and the need for expert labeling. Synthetic data offers a promising solution, with potential impact in critical areas such as lung cancer, the leading cause of cancer-related deaths [3]. Diffusion models [11] have emerged as the most powerful generative framework, surpassing VAEs [16] and GANs [7] in realism and stability [5, 19]. However, scaling diffusion models to large synthetic volumes such as CT scans remains challenging due to extreme computational demands [14]. Recent methods have explored efficiency trade-offs. Previous Latent Diffusion Models (LDMs) [22] for 3D synthesis use autoencoders for data compression, but are often limited in resolution [21, 15]. PatchDDM [2] and WDM [6] bypass autoencoders with subvolume or wavelet representations but still require large GPU memory. NVIDIA's LDM MAISI [8] attains the highest resolution to date ($512 \times 512 \times 768$), but demands 49.7GB GPU memory, unaffordable for most users.

We introduce LAND (Lung-And-Nodule-Diffusion), a memory-efficient latent diffusion model for 3D chest CT synthesis. It generates $256^3$ volumes at 1mm resolution on a single 20GB GPU, uses lung and nodule masks for anatomical conditioning, and controls nodule texture for realistic pathological diversity. LAND combines computational efficiency with fine-grained anatomical control to achieve state-of-the-art (SOTA) high-resolution volume synthesis with practical hardware requirements.

## 2 Method

LAND is a latent diffusion model comprising a 3D U-Net and a 3D VAE architecture (Fig. 1).

**3D VAE** A 3D VAE encodes input CT images into latent representations compressing $4\times$ the spatial resolution and expanding $4\times$ the feature dimensionality: each $256 \times 256 \times 256$ volume

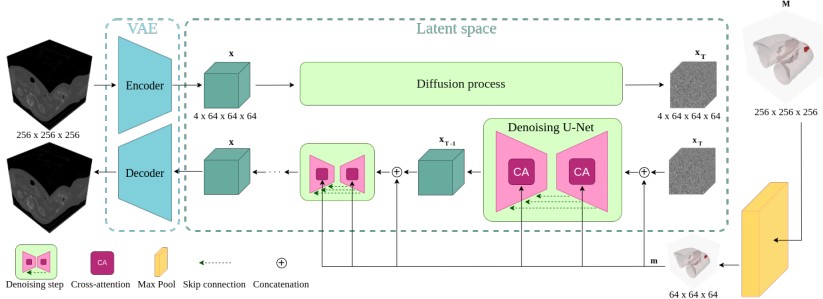

Figure 1: LAND uses a 3D VAE to encode CT volumes into a latent space, where a 3D U-Net performs diffusion on the latent samples $\mathbf{x}$, optionally conditioned on anatomical masks $\mathbf{m}$ (lungs/nodules).

is encoded as a $64 \times 64 \times 64 \times 4$ latent sample. We adopt a lightweight variant of the MAISI architecture [8], using 3 resolution levels with one residual block per level and the same number of channels in both encoder and decoder. The VAE is trained using a combination of an $L_1$ loss $\mathcal{L}_{\mathrm{MAE}}$, a perceptual similarity loss $\mathcal{L}_{\mathrm{LPIPS}}$, an adversarial loss $\mathcal{L}_{\mathrm{ADV}}$ and a Kullback-Leibler term $\mathcal{L}_{\mathrm{KL}}$: $\mathcal{L}_{\mathrm{VAE}} = \mathcal{L}_{\mathrm{MAE}}(\mathbf{x}, \hat{\mathbf{x}}) + \mathcal{L}_{\mathrm{LPIPS}}(\mathbf{x}, \hat{\mathbf{x}}) + \mathcal{L}_{\mathrm{ADV}}(\mathbf{x}, \hat{\mathbf{x}}) + \mathcal{L}_{\mathrm{KL}}(\mathcal{E}(\mathbf{x}))$, where $\hat{\mathbf{x}} = \mathcal{D}(\mathcal{E}(\mathbf{x}))$ is the reconstructed encoded-decoded volume. $\mathcal{L}_{\mathrm{MAE}}$ and $\mathcal{L}_{\mathrm{LPIPS}}$ enforce numerical and perceptual fidelity [25], while $\mathcal{L}_{\mathrm{KL}}$ regularizes the latent space [16] and $\mathcal{L}_{\mathrm{ADV}}$ prevents unrealistic artifacts [8, 7].

**3D U-Net** The denoising network is a 3D U-Net with 5 resolution levels and two residual blocks per level. Additive skip connections [2] reduce memory load while preserving spatial information. To enhance conditional generation, cross-attention modules [22, 17] re-inject conditioning masks (if any) at multiple resolution levels. Training uses velocity prediction [23], enabling the U-Net to learn denoising by estimating a linear combination of clean latent and added noise, which stabilizes training and improves high-resolution synthesis [12, 23]. A linear noise schedule is applied, and training follows a Min-SNR-$\gamma$ loss weighting [9] to balance timestep contributions by signal-to-noise ratio: $\mathcal{L}_{\mathrm{min\text{-}SNR}} = \gamma(\mathrm{SNR}_t)|\hat{\mathbf{v}}_t(\mathbf{z}_t, \mathbf{m}) - \mathbf{v}_t|^2$, where $\mathbf{z}_t$ is the noisy latent, $\mathbf{m}$ the conditioning mask, $\gamma(\cdot)$ the Min-SNR weight, and $\mathbf{v}_t, \hat{\mathbf{v}}_t$ the target and predicted velocities. To ensure anatomical plausibility in 3D, LAND can be conditioned on masks $\mathbf{m}$ covering lungs and nodules. Unlike prior 2D work [17], where nodule-only masks led to implausible nodule placements, our volumetric setting proposes richer conditioning. Spatial and textural cues are encoded by assigning lungs a value of 0.5 and nodules 1–5 (non-solid to solid). Masks are normalized to [0,1], downsampled four times via 3D max pooling, concatenated with the noisy latent, and injected into U-Net cross-attention layers [22, 17].

## 3 Experimental Results

**Datasets and Evaluation** Two publicly available datasets were used. LIDC-IDRI [1] includes 1,010 CT volumes with nodule masks and attribute ratings from four radiologists; for this study, nodule textures scored 1–5 (1: Non-Solid, 2: Non-Solid/Mixed, 3: Part-Solid, 4: Solid/Mixed, 5: Solid) were considered. From NLST [20], we selected 881 CT volumes with at least one nodule annotation [18] and generated the nodule masks using an ad-hoc U-Net. Lung regions in both datasets were segmented with a pre-trained open-source U-Net [13]. All scans were preprocessed as in [6]. LIDC-IDRI was used for training, while the NLST subset provided unseen anatomical masks for inference. Evaluation follows the protocol of previous SOTA models [6], using Fréchet Inception Distance (FID) [10] for synthesis quality and MS-SSIM [24] for sample diversity. FID is computed on 881 real and synthetic CT scans using a ResNet-50 pretrained on 23 medical imaging datasets [4]; lower FID indicates closer distributional alignment between real and synthetic samples. MS-SSIM is computed on 10k synthetic pairs, with lower scores indicating higher diversity.

**Implementation Details** Training was performed on a single Nvidia Grid A100-20C (20 GB) GPU. The 3D VAE was trained independently for 100 epochs with AdamW (learning rate $1 \times 10^{-4}$, batch size 1). The 3D U-Net was trained for 500k steps with AdamW (learning rate $1 \times 10^{-5}$, batch size 1). The diffusion process used $T = 1000$ timesteps with a linear noise schedule from $\beta_1 = 1 \times 10^{-4}$ to $\beta_T = 0.02$. Inference uses the same number of steps to prioritize sample quality.

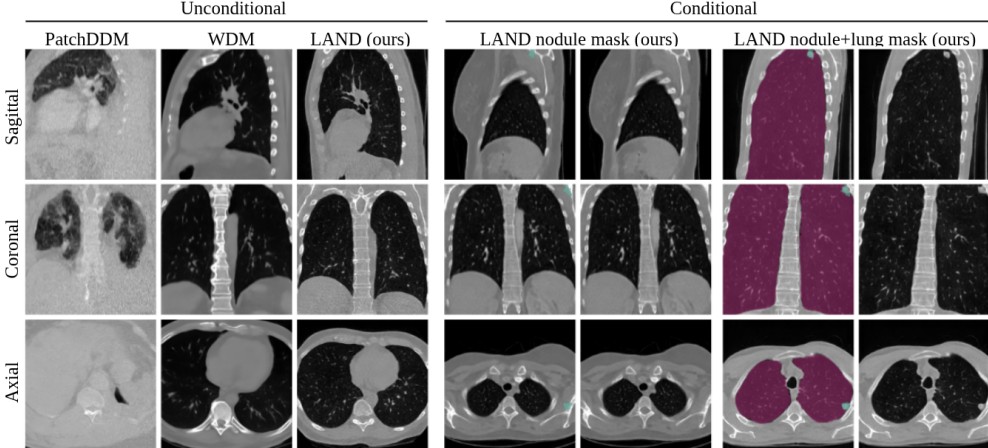

Figure 2: Comparison of unconditional (left) and conditional (right) CT generation using LAND and baseline methods PatchDDM [2] and WDM [6]. Mask overlays are shown where applicable.

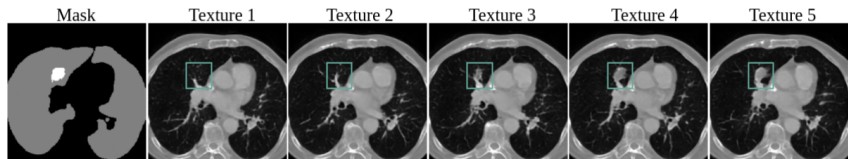

Figure 3: LAND samples conditioned on nodule+lung+texture masks, with increasing texture scores.

**Discussion** We evaluate the unconditional LAND pipeline against WDM [6] and PatchDDM [2], with WDM using the official pre-trained weights and PatchDDM retrained for our task. Quantitatively, LAND achieves the lowest FID (Table 1-Left), reflecting higher fidelity and semantic alignment through its VAE latent space; WDM shows slightly higher MS-SSIM, indicating greater diversity but potentially related to some unrealistic samples, while PatchDDM performs worst, likely due to the inability to handle unregistered volumes. Qualitatively, LAND produces sharp, anatomically consistent samples, WDM tends to appear slightly blurred, and PatchDDM exhibits higher levels of noise and structural variability (Fig.2-Left). Note that LAND and WDM have similar inference memory requirements, but only LAND can be trained on a single 20GB GPU, whereas WDM requires double the memory. Differences in WDM performance compared to [6] likely stem from the differing sample count used for the FID computation, as FID can be sensitive to sample count. Conditional LAND experiments with masks—(1) nodules, (2) nodule+lung, and (3) nodule+lung+texture—show improved FID when lungs are included, highlighting the importance of global context, while MS-SSIM remains similar (Table 1-Right). Only nodule+lung masks (Fig.2-Right) ensure realistic nodule placements. Nodule masks (without lung areas) may incorrectly lead to nodules outside the lungs. Nodule+lung+texture masks further allow control over the synthetic nodule solidity (Fig.3).

Table 1: Comparison of LAND (ours) with SOTA methods. FID values are multiplied by $10^3$.

| Unconditional Method | FID↓ (LIDC) | FID↓ (NLST) | MS-↓ SSIM | Mem↓ (GB) | Conditional Method | FID↓ (LIDC) | FID↓ (NLST) | MS-↓ SSIM | Mem↓ (GB) |
|---|---|---|---|---|---|---|---|---|---|
| PatchDDM [2] | 317.53 | 376.4 | 0.39 | 19.61 | LAND nodule | 4.52 | 5.82 | 0.3 | 7.52 |
| WDM [6] | 15.24 | 32.66 | **0.27** | **7.27** | LAND nodule+lung | **4.48** | **3.37** | 0.29 | 7.52 |
| LAND | **5.062** | **4.76** | 0.29 | 7.38 | LAND nodule+lung+texture | 4.60 | 3.87 | 0.29 | 7.52 |

# 4 Conclusion

This paper presents LAND, a latent diffusion model that generates high-quality chest CT volumes from anatomical masks. The method enables precise control of lung and nodule characteristics while remaining efficient on a single mid-range GPU. Future work includes testing LAND synthetic samples for tasks such as nodule classification and segmentation, extending the model with additional clinically relevant features, and adding a mask generation module to enhance anatomical diversity.

## 5 Potential Negative Societal Impact

While the proposed approach offers useful tools for medical research and education, it also presents potential risks that should be acknowledged. High-quality synthetic 3D chest CT scans could be mistaken for real clinical data if not clearly labeled, which might lead to confusion or reduce trust in medical imaging workflows. There is also a possibility that biases present in the training data could be reflected or amplified in the generated outputs.

The proposed method enhances image quality and lowers computational cost without introducing new misuse risks beyond those already known in generative modeling. Careful data governance, clear labeling of synthetic content, and responsible use are important to minimize unintended negative consequences.

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
