# OpenReview forum: "LAND: Lung and Nodule Diffusion for 3D Chest CT Synthesis with Anatomical Guidance"
_EurIPS.cc/2025/Workshop/MedEurIPS — EurIPS 2025 Workshop MedEurIPS Submission_

### Official Review · Reviewer_zwki · 2025-10-28
**Review comments**

**Rating:** 6
**Confidence:** 5

**Review:**

This paper presents LAND, a conditional generative model for chest CT volume generation from anatomical masks.

The technical contribution is vague and lacks novelty. The claimed reduction in GPU usage is primarily attributed to a straightforward downsampling of input/intermediate tensor size (from 512 to 256), which is not a substantial technical advancement.

The paper is well-written and well-organized. However, the authors must clearly define and elaborate on the detailed technical novelty of the proposed method.

---

### Official Review · Reviewer_BV7i · 2025-10-31
**Review - LAND: Lung and Nodule Diffusion for 3D Chest CT Synthesis with Anatomical Guidance**

**Rating:** 6
**Confidence:** 5

**Review:**

The authors propose LAND, a memory-efficient diffusion model that generates images conditioned on lung nodules and lung masks. The method shows promising results in low-memory settings. It is evaluated qualitatively and using metrics such as FID, SSIM, and MS-SSIM.

Strengths:

- Conditioning with a strong prior, such as nodules and segmentation, is a promising research
- The paper is well written and easy to understand
- Enabling training of diffusion models in low-resource settings helps increase their applicability and is a nice motivaiton.
- Results are tested on two datasets.

Weaknesses:
- The paper is missing a clear take-home message due to the lack of ablations, which makes the contribution weaker. For example, MAISA should be part of the baseline table with proper experiments comparing training cost and generative quality.
- There is no proper check verifying that the generated volumes are purely synthetic. In low-data, high-resolution settings, this should be properly evaluated, especially if the model uses strong conditioning.
- The method uses a pretrained U-Net, which means that the diffusion model and the mask conditioning inherit biases from the pretrained method.
- The evaluation could be stronger. For example, it would help to test whether the generated images can be used for augmentation, imputation, or counterfactual generation.

Questions:
- Did you look into not fine-tuning the 3D VAE but instead using pretrained ones? In my experience, there is no downside to using well-trained pretrained models.

Given the nature of MedNeurIPS and the limited page numbers, I can recommend the paper for acceptance, as I believe it will lead to fruitful discussions.

---

### Decision · Program_Chairs · 2025-10-31

**Decision:**

Accept (Poster)

**Comment:**

Both reviewers find the paper well written and relevant but note limited technical novelty and missing ablations. The idea of anatomically guided diffusion is interesting and suitable for discussion.